# Reactions to macro-level shocks and re-examination of adaptation theory using Big Data

Talita Greyling[1,2☯], Stephanié Rossouw[3,4☯]*

1 School of Economics, College of Business and Economics, University of Johannesburg, Gauteng, South Africa, 2 Honorary Adjunct Academic, School of Social Science & Public Policy, Auckland University of Technology, Auckland, New Zealand, 3 School of Social Science & Public Policy, Faculty of Culture and Society, Auckland University of Technology, Auckland, New Zealand, 4 School of Economics, College of Business and Economics, University of Johannesburg, Johannesburg, South Africa

☯ These authors contributed equally to this work.
* stephanie.rossouw@aut.ac.nz

**Data Availability Statement:** All relevant data are within the paper and its Supporting Information files.

**Funding:** The author(s) received no specific funding for this work.

## Abstract

Since 2020, the world has faced two unprecedented shocks: lockdowns (regulation) and the invasion of Ukraine (war). Although we realise the health and economic effects of these shocks, more research is needed on the effect on happiness and whether the type of shock plays a role. Therefore, in this paper, we determine whether these macro-level shocks affected happiness, how these effects differ, and how long it takes for happiness to adapt to previous levels. The latter will allow us to test whether adaptation theory holds at the macro level. We use a unique dataset of ten countries spanning the Northern and Southern hemispheres derived from tweets extracted in real-time per country. Applying Natural Language Processing, we obtain these tweets' underlying sentiment scores, after which we calculate a happiness score (Gross National Happiness) and derive daily time series data. Our Twitter dataset is combined with Oxford's COVID-19 Government Response Tracker data. Considering the results of the Difference-in-Differences and event studies jointly, we are confident that the shocks led to lower happiness levels, both with the lockdown and the invasion shock. We find that the effect size is significant and that the lockdown shock had a bigger effect than the invasion. Considering both types of shocks, the adaptation to previous happiness levels occurred within two to three weeks. Following our findings of similar behaviour in happiness to both types of shocks, the question of whether other types of shocks will have similar effects is posited. Regardless of the length of the adaptation period, understanding the effects of macro-level shocks on happiness is essential for policymakers, as happiness has a spillover effect on other variables such as production, safety and trust.

## 1. Introduction

Since 2020, the world has faced two unprecedented macro-level shocks. The first was lockdown, which was government-regulated to slow the spread of the coronavirus and protect

**Competing interests:** The authors have declared that no competing interests exist.

human life (see Section 2.1 for a full discussion on lockdown shocks). The second was the invasion of Ukraine by Russia, which had a complete lack of respect for human life. Although there is a widely accepted proposition that macro-level shocks negatively affect health and the economy (Baldwin [1], Ludvigson et al. [2], Lu et al. [3], Fetzer et al. [4]), more research is needed on the effect on people's happiness since these are two shocks with differing intent (protect human life vs disregard human life). By investigating a lockdown and an invasion shock, we will be able to inform policymakers on whether the effect on happiness differs, the severity of the effects and how long it takes people to adapt to these types of shocks.

Whereas macro-level shocks, such as the COVID-19 pandemic, are generally seen as unanticipated and exogenous (Morrison et al. [5]), we note that the lockdown due to government regulation was *somewhat anticipated* and endogenous. After China imposed the first coronavirus lockdown in Wuhan on 23 January 2020, people worldwide speculated whether their governments would follow suit. Regardless of the speculation, no one knew for certain whether they would face lockdown or how stringent the policy would be. However, the *ultimate effect of lockdowns through limiting individuals' freedom and rights was unanticipated*, with a negative impact recorded across emotions and economic and other social factors (see section 2). Similarly, the invasion of Ukraine was an anticipated shock for Ukraine and Russia. Still, the subsequent negative effects which impacted *countries not directly involved in the conflict were unanticipated* and exogenous. Considering the above, we acknowledge that these shocks affected countries differently. In the panel under investigation, the shock of the lockdown *directly* impacted the countries. In contrast, the invasion of Ukraine *indirectly* impacted these countries, as the war did not take place in these countries.

These shocks caused a surge of strong negative emotions such as anger and anxiety. Studies such as Smith et al. [6] and Abadi et al. [7] showed that measures, such as lockdowns, caused considerable tension since people reported feeling angrier, more aggressive and getting into confrontations with others. Likewise, we saw that at the national level, anger led to protests (sometimes violent) against lockdowns and governments across the globe (Belgium, New Zealand, Australia, Canada, and the USA). Similarly, we saw people take to the streets against President Putin and his security council when Russia invaded Ukraine in 2022 (nearly 1800 demonstrations between 4 February and 4 March 2022 (ACLED [8]). Additionally, these shocks increased instability in resources, the provision of gas, food and other markets, leading to weaker economic growth, higher unemployment, higher inflation rates and increased poverty.

Previous studies either i) explored adaptation patterns of individuals for one single event at the micro-level, therefore, not necessarily informing us about adaptation patterns across events (Brickman et al. [9]), ii) provided standardised information on how individuals' well-being develops in the approach to and aftermath of major life events relying solely on micro-level survey data (Clark et al. [10], Clark and Georgellis [11], Frijters et al. [12], and Rudolf and Kang [13]) or iii) considered macro-level shocks but primarily focused on *emotions*, *life satisfaction*, *health*, *the economy*, *mental health and loneliness* (Arampatzi et al. [14], Saad [15], Fredrickson et al. [16], Metcalfe et al. [17], Garcia and Rimé [18], Coupe [19], Buntain et al. [20], Clark et al. [21], Welsch [22], Morrison et al. [5], Metzler et al. [23], Rossouw and Greyling [24], Ohrnberger [25], Blanchflower and Bryson [26], Adams-Prassl et al. [27], Hamermesh [28], Brodeur et al. [29], Mahmud and Riley [30], Sibley et al. [31]) (see sections 2.1 and 2.2 for a full discussion).

To this end, our primary aim is to compare two macro-level shocks (lockdown and the invasion of Ukraine) for ten countries spanning the Northern and Southern hemispheres to determine if and how these macro-level shocks affect happiness. Following this, we will compare the effect size of these shocks to gain additional insights. A secondary aim is determining

how quickly countries' happiness levels adapt after experiencing these macro-level shocks. This will, in turn, allow us to determine whether adaptation theory holds at the macro level (similar to studies conducted at the micro level). We compare countries in the Northern and Southern hemispheres to glean additional information and use it as a robustness test for our findings from the whole sample.

To achieve the study's aims, we first conducted Difference-in-Differences (DiD) estimations to establish the relationship between the two macro-level shocks and the observed effect on happiness (GNH). Subsequently, we also conducted a DiD estimation for the Northern and Southern hemispheres to determine if these shocks have a similar relationship with GNH across hemispheres. Second, we perform an event study to test our DiD results' robustness and determine the macro-level shocks' effect on GNH. We consider GNH from 3 weeks pre-event (lockdown and the invasion of Ukraine) to 4 weeks post-event. This enables us to test adaptation theory at a macro level.

Our analyses use a unique dataset, which we construct from tweets extracted in real-time at a country level. Using the aforementioned, we utilise Natural Language Processing to code (score) the tweets' underlying sentiment, and equations are applied to the sentiment scores to derive a happiness score per tweet. We derive the mean daily scores to develop our daily happiness time-series data per country, i.e., the Gross National Happiness (GNH) index. The GNH, our variable of interest, measures well-being/happiness and has been well-established in previous studies (see section 3.2 for a full discussion and also Greyling and Rossouw [32], Greyling et al. [33, 34], Rossouw et al. [35, 36], Morrison et al. [5] and Sarracino et al. [37, 38]).

Our results from the DiD show a strong relationship between the lockdown and invasion of Ukraine shocks and happiness. Subsequently, these results were confirmed by our event study. Therefore, we are confident that our results show that the lockdown and invasion of Ukraine led to a significant *decrease in happiness*. However, when it comes to the magnitude of the shocks on happiness, we found that the lockdown shock was bigger than the invasion shock. Regarding adaptation of happiness levels, we note that adaptation occurred in the second-week post-lockdown and the third-week post-invasion. Therefore, we find compelling evidence that there is a return to a baseline level of happiness for both events, regardless of the type of shock.

Regarding the comparison between the hemispheres, we find similar results to the whole sample. Interpreting the results of the DiD and the event study together, we are confident that the shocks caused a decrease in GNH, thereby confirming the robustness of our initial DiD and event study results. Regarding adaptation, we find that compared to the week of the lockdown, GNH levels pre-event and again in week two post-event (Northern hemisphere) and week one post-event (Southern hemisphere) were significant and higher than the event week itself. It signals a slightly faster adaptation in the South. Structural differences between the Northern and Southern hemispheres could be responsible for the faster adaptation to previous happiness levels in the South, such as school holidays. Regarding the Ukraine invasion, the Southern hemisphere has results similar to the analysis for all the countries. However, we still find that adaptation was somewhat faster for the Southern hemisphere, with GNH levels positive and significant by week three post-event compared to week four for the Northern hemisphere. We find this plausible since proximity and spatial distribution play vital roles in determining the effect of a disaster on happiness (Rehdanz et al., [39]).

Following the above, our study makes several contributions to the existing literature. First, no other study has investigated two different types of macro-level shocks, such as a lockdown and war (the invasion of Ukraine), on *happiness at a macro-level*. Second, no study has yet tested whether adaptation theory holds at the country level across events using a cross-country

analysis. Third, this is the first study to use real-time data sourced from Big Data instead of survey data to re-examine adaptation theory.

The rest of the paper is structured as follows. The next section discusses studies related to adaptation theory and shocks. Section 3 describes the data and the selected variables and outlines the methodologies used. The results and discussion follow in section 4, while the paper concludes in section 5.

## 2. Literature review

The study focuses on determining if and how macro-level shocks affect happiness as well as the length of the adaptation period. Therefore, our literature review discusses studies investigating the effect of macro-level shocks on life satisfaction and emotions (since studies focusing on happiness are rare) and adaptation theory at the micro-level (since evidence from macro-level studies is scarce).

### 2.1 Macro-level shocks

Studies that investigated macro-level shocks range from natural disasters (Rehdanz et al. [39]), economic bailouts (Arampatzi et al. [14]), conflict, terrorism and war (Saad [15], Fredrickson et al. [16], Metcalfe et al. [17], Garcia and Rimé [18], Coupe [19], Buntain et al. [20], Clark et al. [21], Welsch [22]), and COVID-19 and lockdown (Morrison et al. [5], Metzler et al. [23], Rossouw and Greyling [24], Ohrnberger [25], Blanchflower and Bryson [26], Adams-Prassl et al. [27], Hamermesh [28], Brodeur et al. [29], Mahmud and Riley [30], Sibley et al. [31]).

Regarding natural disasters, Rehdanz et al. [39] focused on how a tsunami and nuclear accident at Fukushima in Japan impacted people's *SWB*. Using survey data, they found that proximity and spatial distribution played key roles–the closer people lived to nuclear power stations, the bigger the drop in happiness. Arampatzi et al. [14] studied the *SWB* impact of stress and anxiety about the macro-level shock of the Greek bailout referendum in 2015. Using survey data collected from university students, they found that SWB levels were impacted significantly. Furthermore, they found that individuals with more positive expectations before the referendum announcement experienced smaller decreases in SWB, and they adapted more quickly to this macro-level shock than individuals who held negative expectations regarding the future.

Saad [15] and Fredrickson et al. [16] studied people's *emotional responses* following the 11 September 2001 terrorist attacks in the United States (U.S.). Saad [15] found that the primary emotion felt was anger, followed by sadness and disbelief. Fredrickson et al. [16] found that U. S. college students who exhibited positive emotions pre-attack coped better despite experiencing shock. Metcalfe et al. [17] was the first study to present causal evidence of an international spillover effect of terrorism using people's *SWB* from countries not directly impacted by 9/11. They found a significant and negative effect on the United Kingdom's *mental well-being*.

Garcia and Rimé [18] and Coupe [19] focused on the Paris terrorist attacks of November 2015. Although Garcia and Rime [18] used Big Data in the form of 62,000 Twitter users, their main interest lay in *collective emotions*. They found a collective negative emotional response and a long-term increase in solidarity-related lexical indicators. Expressions of social processes, prosocial behaviour, and positive affect were higher in the months after the attacks for the individuals who participated to a higher degree in the collective emotion. Coupe [19] focused on the French's *mood, expectations and trust*. The author found evidence of increased trust in the national government and reduced optimism. However, there was no evidence that current life satisfaction or political orientation was affected.

Buntain et al. [20] also used Big Data to investigate the *mood* among Twitter users after the Boston Marathon Bombing (BMB) on 15 April 2013. After analysing approximately 134 million tweets, they found a significant increase in using the word 'fear'. Clark et al. [21] investigated the negative effect of the BMB on experienced well-being. They found the effect size to be 0.79 points, corresponding to a one-third of a standard deviation decrease.

In a study of 44 countries actively involved in civil conflict and using average *happiness* by country from representative surveys (World Database of Happiness from Veenhoven [40]), Welsch [22] found that the current number of conflicts significantly reduced the well-being of people in civil wars. However, Welsch [22] concluded that the change in the number of victims (the number of victims per 1000 inhabitants), rather than their absolute number, reduces well-being in civil wars, suggesting that there is some, but no complete, adaptation to the conditions of a war shock. Additionally, he calculated that, on average, the compensating variation for one fatality is about 108,000 US dollars. This means that income must increase by 108,000 US dollars to have happiness at constant levels when one additional person dies. The direct effects of suffering, fear and agony are larger than the indirect effects due to the smaller income brought about by premature death. The idea that some adaption occurs could be that people get used to the horrors of war to some extent. When one experiences that many persons die, the fact that one's son, husband, father or other relative has died may be more bearable because the people affected know that they are no exception to the rule. Other persons have had to come to terms with similar grief. On the other hand, one can reasonably argue that experiencing the death of many others has a cumulative negative effect on one's own happiness.

Morrison et al. [5] used the *GNH* and Plutchik's [41] *wheel of emotions* to investigate the response to the COVID-19 pandemic in New Zealand. The authors found distinct reactions to the pandemic from 14 February to 14 June 2020. There was an initial strong decrease in happiness in response to COVID-19, which was brought on by a decrease in the emotions 'joy', 'anticipation' and 'trust'. However, this initial decrease in happiness was short-lived and recovered relatively quickly. Focusing on the first five weeks after the outbreak of the COVID-19 pandemic in 2020 for 18 countries, Metzler et al. [23] used emotion analyses to investigate *anger*, *sadness*, *anxiety and positive sentiment*. Tweets containing sadness and anger rose gradually and remained close to their highest level until the end of the investigation period. In contrast, anxiety showed a stronger and more immediate increase than sadness and anger but decreased again before the end of the investigation period. Interestingly, they also found that positive sentiment did not significantly differ from its baseline in 2019 and remained relatively stable (Rossouw and Greyling [24] also found joy remains during shocks).

Lastly, when it comes to the macro-level lockdown shock, Ohrnberger [25], Blanchflower and Bryson [26], Adams-Prassl et al. [27], Hamermesh [28], Brodeur et al. [29], Mahmud and Riley [30] and Sibley et al. [31] all successfully showed the negative impact of lockdown shock on, for example, *health*, *the economy*, *mental health and loneliness*. Ohrnberger [25] used difference-in-differences estimations on a longitudinal sample of 6,437 South Africans to investigate the causal effect of the lockdown shock on health and found that it significantly reduced health by 0.2 standard deviations. Blanchflower and Bryson [26] discussed how the US's economic shock occurred in response to policy efforts to contain the COVID-19 virus by going into lockdown in March 2020. Adams-Prassl et al. [27] used real-time US survey data to compare mental health before and after the introduction by state of lockdowns. They found that lockdown decreased mental health by 0.083 standard deviations and that changes in women's mental health mainly drove it. Hamermesh [28] examined data from the American Time Use Survey for 2012–2013 and used those data to simulate what would happen in lockdowns, given that this would result in people staying at home. He predicted that single people were less satisfied with life in running simulations than married people.

Brodeur et al. [29] used Google Trends data to test whether the lockdowns implemented in Europe and the US led to changes in well-being-related topic search terms. Their results suggest that the lockdown may have severely affected people's mental health. Mahmud and Riley [30] provided evidence of the impact of the COVID-19 lockdown shock on the economic and well-being of a sample of households in rural Uganda. Their results were among the first to show a significant negative impact of the lockdown on a rural population. Sibley et al. [31] found that lockdown regulations slightly increased people's sense of community and trust in institutions. However, on the other hand, they cautioned that there would be longer-term challenges to mental health since anxiety and depression levels were up post-lockdown.

## 2.2 Adaptation theory

The literature on adaptation theory is vast; however, we limit our discussion to a few seminal and across-life events studies at a micro-level that investigated adaptation.

Subjective well-being (SWB) research has been significantly affected by the hedonic treadmill theory (see, for example, Mancini et al. [42], Diener et al. [43], Kahneman et al. [44]). Lykken and Tellegen [45] used the hedonic treadmill theory to conclude that adaptation is quick, complete, and inevitable and that genetic predispositions and personality rather than life events accounted for nearly all of the long-term stable variance in SWB. According to this idea of a hedonic treadmill, people have happiness 'set-points' to which they inevitably return following major life events (Brickman & Campbell [46], Headey & Wearing [47], Larsen [48], Williams & Thompson [49]).

However, studies conducted by Diener et al. [43], Lucas et al. [50, 51], although not refuting the happiness set-point model, concluded that even though happiness levels are relatively stable over time, this stability does not preclude large and lasting changes. "*Happiness levels do change, adaptation is not inevitable, and life events do matter*" (Lucas [50: p 78]).

Most evidence for adaptation comes from single-life event cross-sectional studies. Brickman et al. [9] investigated the SWB of people with spinal cord injuries and lottery winners in one of the most cited studies. They found that individuals with spinal cord injuries were happier than expected, and lottery winners were less happy than expected. However, due to limitations such as the lottery winners being marginally happier than the control group (but not significantly happier), individuals with spinal cord injuries going against the adaptation theory expectations (by being significantly less happy than the comparison group, even if they were not quite as unhappy as they expected) and not using longitudinal data to compare pre-and post-events Brickman et al. [9] could not offer strong support for adaptation.

Subsequent studies provide stronger evidence that some form of hedonic adaptation does occur after improving on Brickman et al.'s [9] research design. Studies such as Clark et al. [10], Clark and Georgellis [11], Frijters et al. [12], and Rudolf and Kang [13] went further than single-life event studies such as Brickman et al. [9] by providing standardised information on how individuals' well-being develops in the approach to and aftermath of major life events. Clark et al. [10] and Clark and Georgellis [11] use unemployment, marriage, divorce, widowhood, the birth of a child, and layoff. Frijters et al. [12] use marriage, divorce, childbirth, injury/ illness, death of spouse or child, being a victim of crime, redundancy, change in financial situation, and change in residence. Rudolf and Kang [13] use marriage, divorce, widowhood, unemployment, first job entry, and the shift from the six- to the five-day working week. Using survey data collected at the micro-level from KLIPS, HILDA, GSOEP and BHPS, all the abovementioned studies suggest that the adaptation phenomenon may be general. These studies surmise that people initially react strongly to both positive and negative events, but then their emotional reactions diminish. People return to a positive rather than a neutral happiness

baseline, and life circumstances are necessary to understand long-term SWB; happiness is not completely determined by personality.

## 3. Data

### 3.1 Data and countries

In the analyses, we use two macro-level shocks, namely the first lockdown during COVID-19 and Ukraine's invasion, and compare the pre- and post-event days. The lockdown shock was in 2020, and the invasion in 2022. In choosing our counterfactual period, we follow the pioneering work of Bennet [52], Lewis [53–55], and Menzies and Beebee [56]. In their work, they state that a counterfactual should be closer to actuality than another if the first resembles the actual world more than the second does. Therefore, as a counterfactual period, we use the same days in the year 2021. Another option was using 2019, but 2019 was not 'closer to the actuality' of 2020 and 2022. Simply put, 2021 is a better counterfactual than 2019 because 2021 shares more commonalities with the years of the shocks than 2019. For example, during 2019, COVID-19 or COVID-19-related shocks, such as lockdowns, which had further socio-economic consequences, did not exist. Using 2019 as a counterfactual is not an option since it could not resemble the actual world of 2020 and 2022 because COVID-19 changed the world, and we can never revert back to the times before COVID-19. We assume that 2021 reflects the 'new normal' in which people have accepted COVID-19 as a reality and part of their daily lives.

To compare the data across events for 2020, 2022 and the counterfactual year 2021, we created a day counter with the day of the event being zero. Please note that in subsequent analysis, we refer to these time periods according to the years in which the events happened and the counterfactual year, meaning 2020 refers to the lockdown, 2022 to the invasion, and 2021 to the counterfactual year.

Our analyses include the following ten countries: seven Northern hemisphere countries: Belgium, Germany, Great Britain, France, Italy, the Netherlands, and Spain; and three Southern hemisphere countries: Australia, New Zealand and South Africa. Primarily, the choice of countries is determined by data availability. However, future studies can extend the dataset to include more countries. The current selection of countries from both hemispheres provides unique insights into the effect of two macro-level shocks on happiness and the time needed to adapt. Table 1 summarises key statistics for each country used in the current study.

### 3.2 Twitter data: *GNH.today project*

The outcome variable, the Gross National Happiness (GNH) index, which measures well-being, was sourced from the *GNH.today project*, launched in April 2019 (Greyling et al. [57]). This project measures the evaluative mood of a country's citizens over time. As a measure of mood, the GNH captures the more volatile part of well-being, generally referred to as happiness (Diener et al. [43]). However, the evaluative qualification indicates that tweets reflect individuals' conscious decisions—they evaluate what they want to say. Although the GNH has been well-established and validated in previous studies (specifically see Sarracino et al. [37, 38]), we also discuss the construction of the data here (see also Greyling and Rossouw [32], Greyling et al. [33, 34]), Rossouw et al. [35, 36], Morrison et al. [5]).

Greyling et al. [57] used the Twitter API (v1) to derive the GNH by extracting and harvesting all original tweets within a geographic bounding box corresponding to the country in question. They then perform the necessary data pre-processing after using Google Translate and Microsoft Azure to translate into English languages unavailable in our lexicons, i.e., a library of words that stores the meanings and structures of words. All punctuation, symbols (@, #),

Table 1. Key summary facts of countries in this study.

| Country | Total population | Average happiness levels** (2020) | Date of announcement | Date of the first lockdown (2020) | Date of Ukraine invasion (2022) |
|---|---|---|---|---|---|
| Australia | 25.5 million | 7.09 | 15 March | 17 March* | 24 February |
| Belgium | 11.6 million | 6.98 | 17 March | 18 March | 24 February |
| France | 66.99 million | 6.66 | 16 March | 17 March | 24 February |
| Germany | 83.02 million | 7.08 | 22 March | 22 March | 24 February |
| Great Britain | 66.65 million | 7.17 | 23 March | 23 March | 24 February |
| Italy | 60.36 million | 6.39 | 8 March | 9 March | 24 February |
| Netherlands | 17.28 million | 7.73 | 14 March | 15 March¶ | 24 February |
| New Zealand | 5.5 million | 7.14 | 23 March | 26 March | 24 February |
| South Africa | 57.7 million | 6.32 | 23 March | 27 March | 24 February |
| Spain | 46.94 million | 6.40 | 13 March | 14 March | 24 February |

\* Australia never officially went into a complete lockdown like that in other countries. We used the day when the closure of international borders was announced as a proxy for "lockdown."

¶ The Netherlands started a so-called 'intelligent lockdown' on this date.

\*\* The happiness scores cited here reflect the average for the period in 2020 before the first COVID-19 case was announced.

Source: Greyling et al. [57], Hale et al. [58].

the letters "https", control characters and digits were removed. Natural Language Processing (NLP) performs sentiment analysis and scores each tweet's underlying sentiment. Sentiment analysis is an automated process to determine the feelings and attitudes of the author of a written text (or tweet) (Rossouw and Greyling [59]).

An algorithm drives sentiment analysis and is better than text analysis since it helps you understand an entire opinion and not merely a word from the text. For our sentiment analysis, we use the Sentiment140 lexicon and test the robustness of the coding using other lexicons (see below). Subsequently, every tweet is labelled as having either a positive, neutral or negative sentiment.

After classifying each tweet, we use an equation to derive a net happiness score. The scale of the happiness scores is between 0 and 10, with 5 being neutral; thus, it is neither happy nor unhappy. The index is available live on the *GNH.today project* website (Greyling et al. [57]). To derive time-series data, we calculate the mean GNH per day. In our analyses, we smooth the GNH data to adjust for trends using a 7-day moving average (Kelly [60], Helliwell & Wang [61]).

As a robustness check, we recalculate the GNH using the sentiment scores from the lexicons National Research Council Canada (NRC) (Turney and Mohammad [62]) and Valence Aware Dictionary for Sentiment Reasoning (VADER) (Hutto & Gilbert [63]). If the indices using different lexicons are highly correlated, we assume that the GNH is not sensitive to the type of lexicon used. The calculated GNH using Sentiment140, NRC, and VADER are highly correlated. The Pearson correlation coefficient between GNH (Sentiment140) and GNH (NRC) is $r = 0.88$ ($p = 0.000$) and between GNH (Sentiment140) and GNH (VADER) $r = 0.82$ ($p = 0.000$). Therefore, we assume the GNH index is not sensitive to the type of lexicon used, and the results gained using the GNH index are robust. Greyling et al. [57] conducted internal validity tests to variations in the algorithms used, volumes of tweets, and the sample period, with results presented in the S1 Appendix.

Sarracino et al. [37] tested the external validity of the GNH by checking whether GNH significantly correlates with external data that represent the same or similar concepts. To assess the external convergent validity, they compare cross-sectional country rankings produced by

the GNH to those available from alternative measures of well-being, such as the Eurobarometer's life satisfaction and the World Happiness Report (Helliwell et al. [64]). Additionally, Sarracino et al. [37] analyse the time-series correlation between GNH changes and changes in consumer confidence and other well-being indicators available from Google Trends for the same period. The results are encouraging since the GNH-based country rankings are similar to those based on alternative indicators. Furthermore, the GNH performs relatively well in relation to an index of negative emotions and consumer confidence (Sarracino et al., [37]) (see S2 Appendix).

In addition to the internal and external validity tests conducted, Greyling et al. [34] showed a negative and statistically significant association between the GNH and 'depression' and 'anxiety' in Australia, New Zealand and South Africa. Moreover, we note from Rossouw and Greyling [59] that the GNH accurately captures the mood of a nation (see the study for country examples such as sporting events and the COVID-19 pandemic). Data from South Africa show that the GNH dropped well below previous daily averages following the outbreak of COVID-19. Later, when distancing regulations were implemented, the GNH recovered slightly but remained lower than normal (Greyling et al. [34]).

Working with Big Data in the form of tweets presents benefits such as an abundance of data, heterogeneous users, accounting for the moods of a vast blend of users, providing timely and internationally comparable data, zero non-response bias and the ability to "listen" and observe what people deem important in their lives. Nevertheless, Twitter data have limitations. One of these is that younger individuals are relatively more likely than older individuals to tweet; however, on Twitter accounts, the ages are spread from 19–65, similar to survey data. Another limitation is that we cannot look at the heterogeneous effects of the lockdown and the invasion of Ukraine by demographic groups based on regions. Our results should, therefore, be interpreted as the average impact of lockdown and the invasion on happiness for Twitter users per geographical region. Although we cannot assume that Twitter users are representative of a country, we know that a vast number of the population does have Twitter accounts, and the number of tweets approximates millions per day. Therefore, biases are limited by the size of the sample.

### 3.3 Oxford data and covariates

To select the covariates included in the models, we are limited in our choice of variables, as we can only include high-frequency close to real-time data. Furthermore, the time periods we compare are relatively short. For example, the invasion in Ukraine was in February 2022, and even using real-time data, the number of observations is limited, seeing that the project commenced in April 2022, thus allowing us to use data until the end of April. Given the relatively short period, we are restricted in the number of covariates that can be included in the estimations to avoid overfitting the models. Therefore, we limit our selection of covariates, similar to Fang et al. [65] and Brodeur et al. [29], to the following:

1. The lockdown or invasion variable is the assumed date of the macro-level shock (intervention variable). We construct a dummy variable with 0 before the lockdown or invasion and 1 thereafter. For the lockdown shock, the lockdown date is specific to each country (see Table 1). The invasion date for the Ukrainian war's macro-level shock was 24 February 2022. However, in our analyses, we use the announcement date of the lockdown (see Table 1) since literature has shown that well-being measures react to expectations rather than the event itself (see Brodeur et al. [29], Greyling et al. [34], Morrison [5]). In subsequent analyses, where we compare the Northern and Southern hemispheres, we construct the lockdown and invasion dummy variable following the same method explained above.

We also run all regressions using each country's lockdown dates and the day preceding the invasion as a robustness test.

2. The number of new COVID-19 cases. We use the lagged number of new COVID-19 cases per million to control for the evolution of the pandemic (Hale et al. [58]). Since lockdown is the shock we are investigating, it is imperative that we incorporate a COVID-19-related variable as a control, given the intricacies associated with isolating the unique impact of lockdown on well-being. A failure to control for the effects of COVID-19 could undermine the validity of attributing observed effects solely to lockdown measures rather than the broader influence of the COVID-19 pandemic itself. Here, we follow the work of Rossouw et al. [35], Brodeur et al. [29], Greyling et al. [33, 34] and Sarracino et al. [37], who included COVID-19-related variables as controls.

3. Country fixed effects. We use country fixed-effects to control for countries' observed and unobserved characteristics.

4. A day-of-the-week fixed-effects. We control for any observed or unobserved day-of-the-week effects. For example, Mondays, the first workday of the week, often have lower happiness levels than other days. In contrast, Fridays and Saturdays, in expectation of more leisure time, have higher happiness levels.

5. Season fixed-effects. We control for unobserved factors such as weather.

6. Event-week. We transformed the daily data into weekly data. Week 0 is our event week and our reference week.

## 3.4 Methodology

**3.4.1 Difference-in-Differences.**   We use a Difference-in-Differences (DiD) estimation to investigate the relationship between the two macro-level shocks (lockdown and the invasion) and happiness. The DiD estimation compared GNH for pre- and post-event in 2020 (lockdown) and 2022 (invasion) to the same time periods (counterfactual) in 2021, which is closer to the actuality of 2020 and 2022 (see section 3.1). As mentioned in section 3.3, the lockdown date in our analysis is the date at which the lockdown was announced (Table 1), not the implementation date, as well-being measures react to expectations rather than the event itself (see Brodeur et al. [29], Greyling et al. [34], Morrison [5]). Specifically, we estimate the following equation:

$$GNH_{i,c} = \alpha_0 + \alpha_1 GNH_{i-1,c} + \alpha_2 \eta_{i,c} * Year_i + \alpha_3 X_{i-1,c} + S_i + \mu_c + \sigma_i + \epsilon_{i,c} \qquad (1)$$

Where $GNH_{i,c}$ is the daily happiness at time $i$ for country $c$. $\eta_{i,c}$ is a dummy variable indicating either $lockdown_{i,c}$ or $invasion_{i,c}$ depending on the type of shock, lockdown (2020) or the invasion of Ukraine (2022). The dummy variables $lockdown_{i,c}$ and $invasion_{i,c}$ take on the value of 0 pre-shock (pre-lockdown or invasion) and one after the initial day of post-lockdown or invasion in both the year of the actual lockdown (2020) or war (2022) and the counterfactual year (2021). We report robust standard errors to address heteroscedasticity ($\epsilon_{i,c}$).

$Year_i$ is a dummy variable taking the value of 1 if the year is either 2020 or 2022 and taking the value of 0 for the year 2021. We control for new COVID-19 cases per million with a one-day lag ($X_{i-1,c}$) and a season-fixed effect ($S_i$) to account for unobserved factors such as the weather. Furthermore, the model includes country and day-of-the-week fixed effects recognising the heterogeneity between different days of the week, with Monday normally low and Saturday normally high ($\mu_c$ and $\sigma_i$). Our interaction term $\alpha_1 \eta_{i,c} * Year_i$ will convey the causal impact of the lockdown or invasion on happiness.

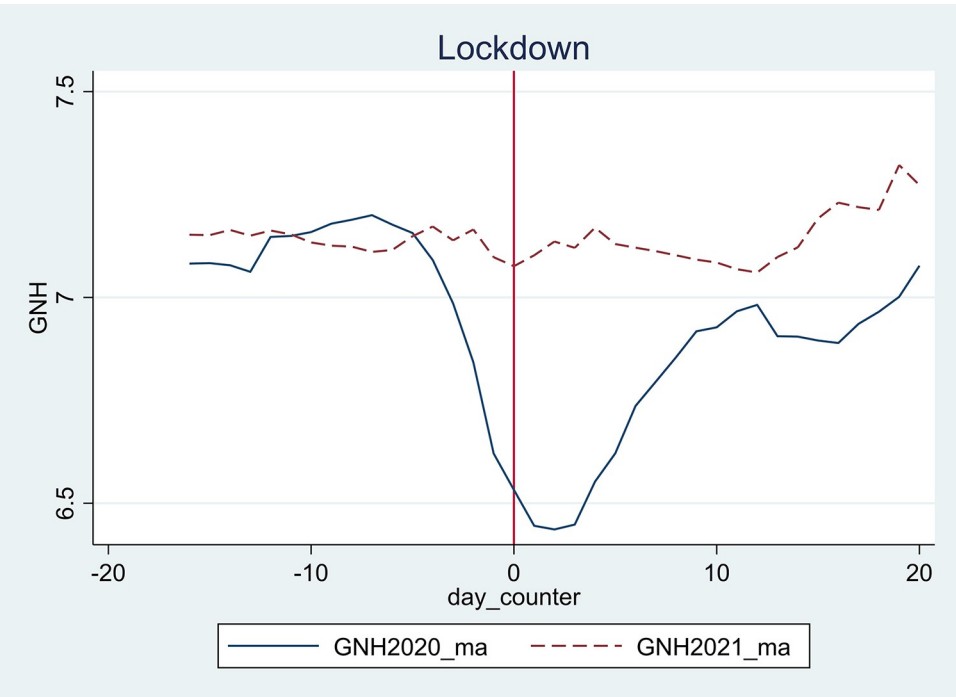

**Fig 1. GNH before and after lockdown.** Note: The vertical axis shows the GNH before and after the day of the event–the red dotted line indicates 2021 (counterfactual), and the solid blue line is 2020, the year of the macro-level shock. Source: Greyling et al. [57].

Since we are working with a long panel, i.e., a few countries compared to many time periods, we tested for stationarity of the GNH time series using the most appropriate panel unit-root test, Im–Pesaran–Shin (Im et al. [66]). We found the presence of a first-grade unit root. We subsequently introduced a lagged dependent variable ($GNH_{i-1,c}$) in our model (Greene [67]), addressing the initial non-stationarity of the dependent variable.

The identification strategy in Eq (1) relies on the assumption that, in the absence of lockdown and invasion, the GNH will have followed similar trends as in the counterfactual year, thus a common-trend assumption. The first step in establishing the common trends assumption is to conduct a graphical analysis. Here, we plot the pre-lockdown and invasion of Ukraine shocks happiness trends (GNH) for 2020/2021 and 2021/2022, respectively. From Figs 1 and 2 in section 4.1, we notice that though the trends are not perfectly parallel, the trends before the lockdown and invasion are similar. Additionally, we see a clear divergence in trends after the shocks and that the trends seem to have been restored after some time has elapsed. We interpret the DiD estimation results with caution due to the lack of perfect common trends and do not claim causality but rather a strong relationship between the shocks and happiness.

We also use context-specific knowledge to argue that 2021 should be selected as the counterfactual since it is similar to the years of the events. COVID-19 occurred in the selected periods; therefore, we are comparing very similar periods since there was no COVID-19 prior to 2020. This means the parallel trends assumption is likely more credible for 2020/2021 and 2021/2022 than if you were comparing periods quite different from one another, i.e., 2019/2020. Considering 2020/2021 and 2021/2022, we also find many similarities between the time periods, thus supporting the parallel assumption. These similarities include i) the seasons are similar, and ii) many public holidays in the countries are on the same date in 2020 and 2021

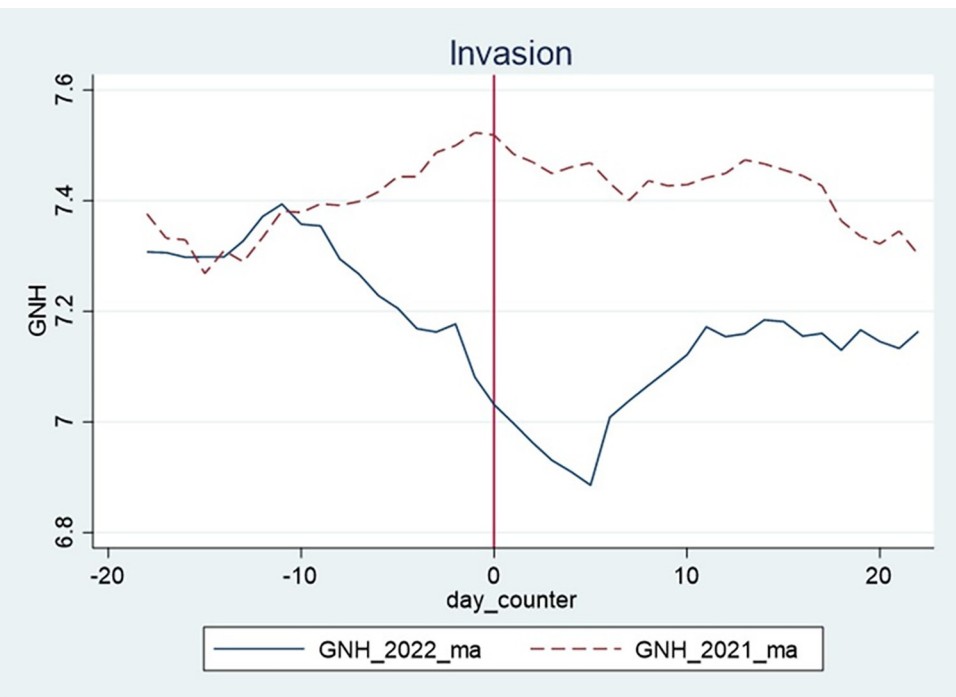

**Fig 2. GNH before and after the invasion.** Note: The vertical axis shows the GNH before and after the day of the event–the red dotted line indicates 2021 (counterfactual), and the solid blue line is 2022, the year of the macro-level shock. Source: Greyling et al. [57].

and in 2022 and 2021 –often capturing higher happiness levels on these days. We also considered other factors, including higher unemployment rates (we saw some recovery late in 2022, but it does not affect our analyses since we analysed the data in February 2022, and as such, the recovery was still not apparent). Other similarities are that GDP growth decreased, inflation rates increased, and consumer spending decreased; these trends were visible in the year of the shocks and the counterfactual year. Furthermore, as mentioned in section 3.1, the choice of the 2021 period makes logical sense. In all time periods, COVID-19 is a reality, not in 2019.

Our identification assumptions are further supported as the dates of the shocks differ across countries; therefore, each country has a unique day 0 (the date of the announcement of the first lockdown). It follows then that if a similar reaction is noticed on the GNH across countries on the day of the announcement, it is most likely due to the shock.

Similarly, the invasion was a unique event, and none of the countries directly participated in the war. Therefore, it is most likely that the observed decrease in GNH across the countries in our sample on the day of the invasion is related to the shock. We also note from the narrative of the extracted tweets that after the lockdown and invasion dates, the tweets related to these events intensified comparatively to 2021, indicating that these events influenced the narrative of people and mattered for their happiness levels, strengthening our case for identification (please also refer to the work done by Brodeur et al. [29]).

Considering the abovementioned, we assume that the macro-level shocks are most likely the cause for the deviation in the GNH from the counterfactual; thus, the probability that these events caused a decrease in happiness is greater than 0, meeting the positivity assumption.

Given that we are working with time series data and the two time series are unlikely to follow the exact same trend, we acknowledge that it is possible the DiD estimator could conflate the true effect of the lockdown and the invasion with the broader negative economic spillover

in 2020 and 2022. Consequently, we are hesitant to draw strict causal conclusions from the DiD and, as a robustness test, perform an event study (see section 3.4.2) to substantiate our findings. Therefore, the results may be interpreted as the average impact of lockdown and the invasion on happiness, comparing pre- and post-lockdown and invasion days in 2020 and 2022 to the same days in 2021 that were assumed to have relatively normal GNH levels.

**3.4.2 Event study.** We estimate an event study model to test for any adaptation of the GNH after the macro-level shocks. Additionally, as mentioned in section 3.4.1, the event study results are a robustness test to confirm the DiD results. For this analysis, we will consider a total of eight weeks. The weeks we will be considering are three weeks pre-event (week -3 to week -1), the week of the event (week 0) and the four weeks after the week of the event (week 1 to week 4).

Therefore, we estimate the following:

$$GNH_{i,c} = \sum_{k=-3}^{4} \alpha_k E_{k,c} + \gamma X_{i-1,c} + \delta GNH_{i-1,c} + S_i + \mu_c + \sigma_i + \epsilon_{i,c} \qquad (2)$$

$GNH_{i,c}$ is the happiness measured at time $i$ for country $c$. $E_{k,c}$ for $k = -3...,4$, are dummy variables for the three weeks pre-event, the event week and four weeks post-event. As with Eq (1), we include lagged GNH ($GNH_{i-1,c}$), and we control for season-fixed effect ($S_i$), country and day-of-the-week fixed effects ($\mu_c$ and $\sigma_i$). The week of the event (week 0) is the reference period. The estimated coefficients on the $E_{k,c}$ dummies should, therefore, be interpreted as the effect of being in (for example) the third-week post-event compared to week 0 (event week).

**3.4.3 Robustness checks.** To test the robustness of our results in the DiD analysis, we also use the lockdown date rather than the announcement date of the lockdown.

Furthermore, as already stated, we use the event study results as a robustness check for our DiD results. If we find similar results from the event study, meaning happiness was negatively affected by the event, we can confirm our DiD results. Lastly, we compare countries in the Northern and Southern hemispheres as a robustness test for our findings from the whole sample and to glean additional information.

# 4. Results

## 4.1 Graphical analysis of the evolution of happiness across events

We start our analysis by graphically comparing the GNH pre- and post-lockdown in 2020 to the same days in 2021. We use line plots, as they clearly show how the GNH changed over time and highlight the salient features captured in the data (Chambers [68]). We investigated the days before and after the lockdown event and followed the same method considering the invasion of Ukraine. However, the reader should note that these graphs only give us an initial picture of what transpired. A more detailed analysis from the DiD and event study estimations, where we control for other factors, should be used to establish whether the changes in the GNH are associated with the shocks.

Fig 1 plots daily GNH for pre- and post-lockdown, and Fig 2 plots GNH for pre- and post-invasion. In both cases, 0 indicates the day of the event (see the X-axis).

Fig 1 shows a sharp decline in GNH (solid blue line) on the day lockdowns were announced and thereafter. This decrease in GNH is only seen in 2020, with no such change in 2021. A similar pattern is seen in Fig 2, with GNH (solid blue line) showing a sharp decrease on the day of the invasion in 2022 (and the days thereafter). However, this is different from the pattern observed in 2021.

The graphical analysis of the GNH data shows a sharp decrease in the GNH brought on by the macro-level shocks. In the absence of lockdown or invasion, we assume the GNH would

have followed a similar pattern as in the counterfactual year, thus a common trend assumption. However, this assumption was violated by the macro-level shocks. The DiD analysis tests this assumption and the relationship between the shocks and GNH.

## 4.2 Difference-in-Differences

To answer our question regarding the relationship between the two macro-level shocks and happiness, we turn to our results from the DiD in Table 2, which compares GNH for the days pre-and post-event in 2020 (lockdown) and 2022 (invasion) to the days (counterfactual) in 2021 for all ten countries, which is closer to the actuality of 2020 and 2022. We restrict our analysis to ten days post-event to get a more robust result to the relationship between the shocks and the effect on GNH noticed from our graphical analyses in section 4.1.

Therefore, to determine whether the decrease in GNH was associated with the lockdown and the invasion specifically and not only an observed trend (since we control for the season, country and day-of-the-week, new COVID-19 cases per million as a proxy for the evolution of the pandemic and previous happiness levels), we consider the estimated coefficient of the interaction variable "$lockdown_{i,c}*Year_{i-1}$ or $invasion_{i,c}*Year_{i+1}$," i.e., the DiD estimator. Table 2 shows that the interaction variable is negative and statistically significant (at the 1% level). Therefore, indicating that "lockdown or the invasion in the year of COVID-19 (2020) and the year of the Ukrainian war (2022)" had significantly lower levels of GNH of 0.111 and 0.057 points, respectively, compared to the same day in the counterfactual year.

We note here that we saw in Figs 1 and 2 that happiness levels started to dip pending the macro-level shocks. The observed dip seems similar to Ashenfelter's dip [69], which posits that individuals approaching an event could already start experiencing changes 'pre-impact' of said event. As explained as part of the event study results in Section 4.3, we note that for the lockdown shock this was likely due to the uncertainty about i) whether a lockdown would be implemented and ii) the effects of a lockdown on all aspects of human life, including social, economic and political dimensions. We argue that the dip pending the invasion shock is plausible since Ukraine was threatened by a Russian invasion months before the invasion. Additionally, the Russian Armed Forces began massing thousands of personnel and military

**Table 2. The effects of COVID-19 lockdown and the invasion of Ukraine—DiD estimates for all countries.**

| Variable | Lockdown | | Invasion | |
|---|---|---|---|---|
| | GNH | SE | GNH | SE |
| Lockdown #year COVID | -0.1115*** | 0.0219 | | |
| Lagged GNH | Yes | | Yes | |
| Lagged new COVID-19 cases per million | Yes | | Yes | |
| Season FE | Yes | | Yes | |
| Country FE | Yes | | Yes | |
| Day of the week FE | Yes | | Yes | |
| Invasion # year War | | | -0.0577*** | 0.0134 |
| _cons | 0.956*** | 0.2135 | 0.3873*** | 0.1813 |
| N | 530 | | 520 | |
| adj. $R^2$ | 0.968 | | 0.970 | |

Standard errors in parentheses

* $p < 0.10$, ** $p < 0.05$, *** $p < 0.01$

Lockdown = Day when lockdowns were announced in each country, and Ukraine = 24 February 2022 (the day of the invasion)

equipment near the Ukraine border and in Crimea, representing the largest mobilisation since the previous invasion in 2014 (refer to Section 4.3 for a full discussion).

The effect size of the decrease in the GNH with the lockdown shock was nearly one-half of a standard deviation, and with the invasion, slightly larger than one-quarter. This is in line with the findings from Clark et al. [21], who found the size of the effect of the Boston Marathon Bombing to be one-third of a standard deviation decrease in experienced well-being. We also notice a lingering effect on the days after the shocks before adaptation to previous happiness levels (see section 4.3).

This leads us to conclude that GNH was significantly affected in both instances, with a high probability that the shocks caused the decreases in GNH.

Thus, in the year when the shocks (treatments) (lockdown and invasion) occurred, people were unhappier on the day of the lockdown announcement and invasion of Ukraine and the days thereafter compared to the counterfactual year. This implies that there is a strong relationship between *the lockdown and the invasion of Ukraine and the decline in happiness levels*, considering previous happiness levels and controlling for trends, seasonality, COVID-19 cases, observed and unobserved differences in the characteristics of countries and day-of-the-week effects. The magnitude of the lockdown shock was bigger than that of the invasion of Ukraine. This is expected as the lockdown directly affected all countries in the sample, whereas the invasion of Ukraine had an indirect effect.

Consequently, we split our sample between the Northern and Southern hemispheres. We see from Table 3 that our interaction variables (year#lockdown and year#invasion) are significant and negatively related to happiness in both hemispheres for both shocks.

These results show that even if we consider the Northern and Southern hemispheres separately, the relationship between *the lockdown and the invasion of Ukraine and the decline in happiness levels* hold for both hemispheres. The fact that the countries in the Southern hemisphere are much further geographically removed from Ukraine makes no significant difference to the effect of the invasion on their happiness levels. The imminent threat of a nuclear attack or World War III and all the other negative consequences, such as supply chain issues and

**Table 3. The effects of COVID-19 lockdown and the invasion of Ukraine–DiD estimates for the Northern and Southern hemispheres.**

| Variable | Lockdown Northern hemisphere | | Lockdown Southern hemisphere | | Invasion Northern hemisphere | | Invasion Southern hemisphere | |
|---|---|---|---|---|---|---|---|---|
| | GNH | SE | GNH | SE | GNH | SE | GNH | SE |
| Lockdown #Year COVID | -0.1131*** | (0.0293) | -0.1275*** | (0.0457) | | | | |
| Lagged GNH | Yes | | Yes | | Yes | | Yes | |
| Lagged new COVID cases per million | Yes | | Yes | | Yes | | Yes | |
| Season FE | Yes | | Yes | | Yes | | Yes | |
| Country FE | Yes | | Yes | | Yes | | Yes | |
| Day of the week FE | Yes | | Yes | | Yes | | Yes | |
| Invasion #Year War | | | | | -0.0599*** | (0.0341) | -0.0513*** | (0.0406) |
| _cons | 1.3868*** | (0.1741) | 1.0860*** | (0.3145) | 0.6516*** | (0.1685) | 0.5491***** | (0.2256) |
| N | 294 | | 126 | | 294 | | 126 | |
| adj. $R^2$ | 0.892 | | 0.895 | | 0.871 | | 0.775 | |

Standard errors in parentheses

* $p < 0.10$, ** $p < 0.05$, *** $p < 0.01$

inflation, affect people's happiness in general, no matter how far they are geographically separated.

## 4.3 Event study

Following the DiD results, we continue with the event study to confirm the DiD results and analyse the period it takes for happiness to adapt after the two macro-level shocks. We divide our time-series data into weeks. The event week is week 0, which includes the announcement of the lockdown dates (see Table 1) and the day of the invasion of Ukraine (24 February 2022). We extend our time period of analysis up to 4 weeks after the event to gain better insights into the adaptation of happiness levels.

From Table 4, we see in terms of the lockdown shock that pre-event GNH levels were significant and higher until the week before the event. However, in the week before the lockdown (week -3), the GNH was negative and significant (although at the 10% level) compared to the event week, which indicates the concern of the population, likely due to higher COVID-19 infection rates and the uncertainty about looming lockdowns. We also note that post the event week, the effect of the shock lingered. However, by week two post-event, the happiness levels are positive and significantly higher than in the week of the shock. For the invasion shock, we see that week -3 had significantly lower happiness levels than the reference week. This is plausible since the USA accused Moscow of a plot to fabricate an attack by Ukrainian forces that Russia could use as a pretext to take military action against its neighbour three weeks before the invasion. Additionally, in week -3, Moscow massed more than 100,000 soldiers near the border with Ukraine. However, the Kremlin denied having plans to attack. This denial seemed to work as weeks -2 and -1 had significantly higher GNH levels than week 0. Weeks one and

**Table 4. Duration of the macro-level shocks on happiness (GNH): lockdown and the invasion.**

| Reference = week 0 (week of the event) | Lockdown | | Ukraine invasion | |
|---|---|---|---|---|
| | GNH_ma | SE | GNH_ma | SE |
| Week -3 | 0.0497*** | (0.0146) | -0.0428*** | (0.0102) |
| Week -2 | 0.0377*** | (0.0130) | 0.0340*** | (0.0074) |
| Week -1 | -0.0817* | (0.0111) | 0.0098** | (0.0074) |
| Week 0 | - | - | - | - |
| Week 1 | 0.0219 | (0.0114) | 0.0255 | (0.0071) |
| Week 2 | 0.0236** | (0.0174) | 0.0292 | (0.0071) |
| Week 3 | 0.0254** | (0.0218) | 0.0350*** | (0.0073) |
| Week 4 | 0.0131 | (0.0218) | 0.0306*** | (0.0076) |
| Lagged GNH | Yes | | Yes | |
| Lagged new COVID-19 cases per million | Yes | | Yes | |
| Season FE | Yes | | Yes | |
| Country FE | Yes | | Yes | |
| Day of the week FE | Yes | | Yes | |
| _cons | 1.1247*** | (0.1181) | 0.4468*** | (0.1016) |
| N | 732 | | 730 | |
| adj. $R^2$ | 0.810 | | 0.698 | |

Standard errors in parentheses

* $p < 0.10$

** $p < 0.05$

*** $p < 0.01$

two post-event show no difference in the level of happiness compared to the event week; therefore, lower happiness levels remained. In week three, happiness levels show adaptation to happiness levels prior to the shock, with a positive and significant value.

Therefore, considering the adaptation period, we find that, even though the lockdown had a bigger effect on happiness than the invasion, the associated adaptation period is shorter for the former compared to the latter. After the lockdown shock, happiness adapted in week two after the shock. Considering the invasion, happiness adapted in week three after the shock, with weeks one and two showing no significant differences from the event week. This result of the invasion shock's adaptation occurring a week later than the lockdown shock is plausible since the invasion created a ripple effect throughout the countries in our study. At first, the people were negatively affected by sadness and sympathy for Ukraine. Still, as time passed, the countries themselves were impacted by rising gas prices, inflation, etc. Therefore, the invasion turned an initial Ukraine problem into an all-country problem, and we only see in week three that the happiness levels were positive and significantly related to the event week (thus adapting to happiness levels before the event week). The above findings, in terms of the length of the shocks' effect on happiness, two weeks (lockdown) and three weeks (Ukraine invasion), are in line with previous research such as Metcalfe et al. [17], Ford et al. [70] and Knudsen et al. [71].

Continuing our investigation into the Northern and Southern hemisphere countries, we refer to the results in Table 5. In both hemispheres, we see the results in terms of the lockdown shock similar to the analysis for all the countries. For the Northern hemisphere, pre-event GNH levels were significant and higher until the week before the event. However, in the week before the lockdown (week -1), the GNH was negative and significant (although only at the 10% level) compared to the event week, which indicates the concern and uncertainty prevalent among the population, likely due to higher COVID-19 infection rates and the uncertainty of what the future will hold. In the event week, the shock of the lockdown announcement was evident, with the two weeks after showing no significant difference (week two only significant at 10%) in the GNH levels compared to the event week; thus, lower happiness levels prevailed. Two weeks after the event (week 3), we see evidence of adaptation to levels of GNH pre-event.

Pre-event GNH levels were higher and statistically significant for the Southern hemisphere up to the week before the event week, when it became no different from the event week. In the Southern hemisphere, the adaptation occurred in the week post-event (compared to two weeks post-event in the Northern hemisphere). Therefore, the Southern hemisphere adapts slightly faster than the Northern hemisphere. Likely reasons for a quicker adaptation after the lockdown shock might be structural differences between the Northern and Southern hemispheres, with the Southern hemispheres moving into winter, their children having school holidays, and therefore being less affected by the lockdown itself. Furthermore, there were promises of only a brief lockdown period.

Regarding the invasion, we note that the Southern hemisphere has results similar to the analysis for all the countries. The GNH levels were higher pre-invasion (except week -3, though not significant) than in the event week, and in the weeks after the invasion, GNH levels were not significantly different from the event week. By week three, adaptation occurred. For the Northern hemisphere, GNH levels were higher and statistically significant pre-invasion (except for week -3; see explanation given for full sample) and positive before the event week (though no longer significant). The effect of the invasion lasted longer in the Northern hemisphere, with happiness levels in week three post-invasion showing to be significantly different from the event week only at the 10 per cent level. Adaptation (GNH significantly higher than the event week) occurred in week four. This lingering effect of lower happiness levels in the Northern hemisphere is plausible since proximity and spatial distribution play vital roles in determining the effect of a disaster on happiness (Rehdanz et al., [39]).

**Table 5. Duration of the macro-level shocks (lockdown and the invasion) on happiness (GNH): Northern vs Southern hemisphere.**

| Reference = week 0 (week of the event) | Lockdown–Northern hemisphere | | Lockdown–Southern hemisphere | | Invasion–Northern hemisphere | | Invasion–Southern hemisphere | |
|---|---|---|---|---|---|---|---|---|
| | GNH_ma | SE | GNH_ma | SE | GNH_ma | SE | GNH_ma | SE |
| Lockdown Week -3 | 0.0359** | (0.0175) | 0.0989** | (0.0481) | | | | |
| Lockdown Week -2 | 0.0289* | (0.0166) | 0.0829** | (0.0333) | | | | |
| Lockdown Week -1 | -0.1080* | (0.0141) | -0.0249 | (0.0273) | | | | |
| Lockdown Week 0 | - | - | - | - | | | | |
| Lockdown Week 1 | 0.0117 | (0.0152) | 0.0768*** | (0.0276) | | | | |
| Lockdown Week 2 | 0.0080* | (0.0228) | 0.1297*** | (0.0413) | | | | |
| Lockdown Week 3 | 0.0131** | (0.0284) | 0.1400*** | (0.0507) | | | | |
| Lockdown Week 4 | 0.0062** | (0.0285) | 0.1143** | (0.0504) | | | | |
| Invasion Week -3 | | | | | -0.0441*** | (0.0123) | -0.0264 | (0.0247) |
| Invasion Week -2 | | | | | 0.0362*** | (0.0087) | 0.0320** | (0.0157) |
| Invasion Week -1 | | | | | 0.0056 | (0.0089) | 0.0300** | (0.0157) |
| Invasion Week 0 | | | | | - | - | - | - |
| Invasion Week 1 | | | | | 0.0277 | (0.0082) | 0.0196 | (0.0156) |
| Invasion Week 2 | | | | | 0.0242 | (0.0083) | 0.0440 | (0.0153) |
| Invasion Week 3 | | | | | 0.0281* | (0.0086) | 0.0596*** | (0.0169) |
| Invasion Week 4 | | | | | 0.0361*** | (0.0093) | 0.0316* | (0.0181) |
| Lagged GNH | Yes | | Yes | | Yes | | Yes | |
| Lagged new COVID-19 cases per million | Yes | | Yes | | Yes | | Yes | |
| Season FE | Yes | | Yes | | Yes | | Yes | |
| Country FE | Yes | | Yes | | Yes | | Yes | |
| Day of the week FE | Yes | | Yes | | Yes | | Yes | |
| _cons | 1.3868*** | (0.1741) | 1.0860*** | (0.3145) | 0.6516*** | (0.1685) | 0.5491** | (0.2648) |
| N | 375 | | 147 | | 364 | | 156 | |
| adj. $R^2$ | 0.792 | | 0.815 | | 0.671 | | 0.782 | |

Standard errors in parentheses

* $p < 0.10$

** $p < 0.05$

*** $p < 0.01$

Therefore, from the event study, GNH levels seem to adapt relatively quickly to previous happiness levels after a shock. This confirms that the adaptation theory holds regarding macro-level shocks, similar to studies focusing on life events at a micro-level as argued by Larsen [48], Lykken and Tellegen [45], Williams and Thompson [49], Headey and Wearing [47] and Brickman and Campbell [46].

Furthermore, interpreting the results of the DiD estimation and the event study together, we are confident that the *shocks led to a decrease in happiness*.

## 5. Conclusions

In this study, we compared two macro-level shocks (lockdown and the invasion of Ukraine) for ten countries spanning the Northern and Southern hemispheres to determine if and how these macro-level shocks affected happiness. Following this, we also compared the magnitude of the effects of these shocks to gain additional insights. We also determined how quickly countries' happiness levels adapted after experiencing these macro-level shocks and therefore tested whether adaptation theory held at the macro level. This also served as a robustness test

for the results of our Difference-in-Differences estimations from the first research question. Lastly, we compared countries in the Northern and Southern hemispheres as a robustness test for our findings from the whole sample and to glean additional information.

By doing the aforementioned, we contributed to the literature in three significant ways. First, no other study compared two different types of macro-level shocks on happiness at a macro level. Second, no other study tested whether adaptation theory holds at the country level across events using a cross-country analysis. Third, this was the first study to use real-time information from Big Data instead of survey data to re-examine adaptation theory. This allowed us to contribute to the debate on whether adaptation at a macro level is quick and complete.

Interestingly, we found similar results *notwithstanding the types of shocks*. Considering our results from the Difference-in-Differences estimations and the event studies, we are confident that the shocks led to lower happiness levels, both with the lockdown and the invasion shock. Additionally, the shocks caused a significant decrease in happiness with effect sizes of nearly one-half (lockdown) and slightly higher than one-quarter (invasion) of the standard deviation of happiness on the day of the shock. Thus, the lockdown had a bigger effect size than the invasion. This finding held for both the Northern and Southern hemisphere countries. Regarding adaptation, we found that whereas happiness levels adapted two weeks after the lockdown event, happiness levels took somewhat longer (three weeks) to adapt after the invasion of Ukraine. We found similar results regarding lockdowns across both hemispheres. However, when it comes to the invasion of Ukraine, the Northern hemisphere's adaptation seems slower compared to the Southern hemisphere.

Therefore, considering both macro-level shocks, we can conclude that happiness levels adapted relatively quickly (three weeks after the event) to previous levels. Our findings are the first to confirm adaptation theory at a macro-level *across events* using a cross-country analysis. Our study is also the first to confirm that the macro-level adaptations of countries to happiness shocks within our sample are similar to the micro-level findings, suggesting that people initially react strongly to negative events. However, their happiness levels revert to previous levels.

It would be negligent of us not to discuss our study's limitations. Our panel of countries under investigation does not include Ukraine, Russia or neighbouring countries. It is plausible that if they had been included, we might not have seen a complete adaptation. Furthermore, as discussed in section 3.2, there are limitations when working with Twitter data. However, despite these limitations, the results contribute to understanding the adaptation to macro-level shocks at a macro level. Future studies should investigate whether the phenomena seen in this paper are also observed considering other macro-level shocks.

## Supporting information

**S1 Appendix. Internal validity of the Gross National Happiness index.**
(DOCX)

**S2 Appendix. External validity of the Gross National Happiness index.**
(DOCX)

**S1 Data.**
(XLSX)

## Author Contributions

**Conceptualization:** Talita Greyling, Stephanié Rossouw.

**Data curation:** Talita Greyling, Stephanié Rossouw.

**Formal analysis:** Talita Greyling, Stephanié Rossouw.

**Investigation:** Talita Greyling, Stephanié Rossouw.

**Methodology:** Talita Greyling, Stephanié Rossouw.

**Project administration:** Talita Greyling, Stephanié Rossouw.

**Resources:** Talita Greyling, Stephanié Rossouw.

**Software:** Talita Greyling, Stephanié Rossouw.

**Validation:** Talita Greyling, Stephanié Rossouw.

**Visualization:** Talita Greyling, Stephanié Rossouw.

**Writing – original draft:** Talita Greyling, Stephanié Rossouw.

**Writing – review & editing:** Talita Greyling, Stephanié Rossouw.

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
