## [Decision Letter · Decision Letter 0]

16 Jan 2023

PONE-D-22-19460Re-examining adaptation theory using Big Data: Reactions to external shocks.PLOS ONE

Dear Dr. Rossouw,

Thank you for submitting your manuscript to PLOS ONE. After careful consideration, I feel that it has merit but does not fully meet PLOS ONE’s publication criteria as it currently stands. Therefore, I invite you to submit a revised version of the manuscript that addresses the points raised during the review process. ***Please note that in light of the reviewers' criticism, I consider this a risky major revision without a guarantee of publication. Therefore, do your best to convince the reviewers, who are experts in your research area, that your revision is at a publishable level, as this might be the last chance for you to do so.***

We look forward to receiving your revised manuscript.

Kind regards,

Ali B. Mahmoud, Ph.D.

Academic Editor

PLOS ONE

Journal Requirements:

2. In your Methods section, please include additional information about your dataset and ensure that you have included a statement specifying whether the collection and analysis method complied with the terms and conditions for the source of the data.

3. We note that you have referenced (Silver, RLet al. [2]) which has currently not yet been accepted for publication. Please remove this from your References and amend this to state in the body of your manuscript: (ie “Silver, RL et al. [Unpublished]”) as detailed online in our guide for authors http://journals.plos.org/plosone/s/submission-guidelines#loc-reference-style

Reviewers' comments:

Reviewer's Responses to Questions

**Comments to the Author**

1. Is the manuscript technically sound, and do the data support the conclusions?

Reviewer #1: Partly

Reviewer #2: Partly

2. Has the statistical analysis been performed appropriately and rigorously? 

Reviewer #1: Yes

Reviewer #2: No

3. Have the authors made all data underlying the findings in their manuscript fully available?

Reviewer #1: No

Reviewer #2: No

4. Is the manuscript presented in an intelligible fashion and written in standard English?

Reviewer #1: Yes

Reviewer #2: Yes

5. Review Comments to the Author

Reviewer #1: 1. The introduction should be rewritten better

1.1 Long introduction and includes few citations. For example, the part between line 55 and line 85 does not have any citation

1.2 The part between line 85 and line 98 is more suitable to be in related works

2. From the text, it is not possible to know if the tweets dataset is suitable for this type of study or not.

Authors should add a table showing how tweets are selected for each country. For example, the language and if the tweet has to contain a specific hashtag.

3. Authors have to add more description about the Ml models that they use to predict the sentiment like:

Are the models trained by the authors or not

is there a model for each language

what is the performance of each model

4. Apart from the factors to be studied, the countries that have been selected are not homogeneous, and each country has different events that affect the level of happiness in it. why It is not mentioned in the text anything about that?

Reviewer #2: The question the authors are addressing is an interesting and important one, and there are features of there data that are attractive – notably their use of Twitter to capture real-time changes in sentiment – but I was not convinced by the empirical analysis and remain uncertain about the contribution to the literature. It’s not entirely clear to me from the descriptions of the data used in the study whether these concerns can be addressed in a revision. I elaborate on these points below.

1. Motivation and Contribution

The motivation and contribution were unclear to me. The authors talk of a contribution at the ‘macro’ level but, in fact, their analysis is really at the micro level using country fixed effects. There is nothing wrong with that – indeed it’s the right way to go empirically – but it means we are focusing on within country variance in sentiment/wellbeing over time, rather than comparing across countries as some might expect given the references to the macro-level in the set up. This probably needs revisiting. As an aside, it was never clear to me why macro effects should differ from an aggregation of the micro-effects in this case. That point also needs discussion if the authors choose to stick to the macro issue as a contribution.

I also thought that the review of existing literature on habituation and adaptation was very partial, missing some important contributions from the literature in conflict, war and terrorism, such as the work of Krueger on 9/11 and the work of Clark et al on the Boston Bomber. These and many other papers have established the rapid return to pre-event wellbeing following horrific events such as bombings. I encourage the authors to go back to that literature and re-read it so as to think more carefully as to what their contribution might be. At present I’m not convinced that contribution is sufficient to merit publication.

2. The characterisation of shocks and the timing of events

The timing of the Ukraine shock as the moment of invasion seems reasonable, notwithstanding some likely anticipation, but the timing of the COVID shock is really the moment the pandemic led to a big increase in infection rates. Yes the infection rate – which arguably captures the intensity of the COVID outbreak and could thus be used to identify before/after periods – is used as a right-hand side control variable in the analyses. This seems odd to me. Instead, the authors focus on the date of lockdowns which are a policy response to the shock, not the shock itself.

If the authors stick with lockdowns as the treatment they need to think hard about whether and why they condition on COVID cases since this nets out much of the health shock. One might want to do that if one is thinking about the effects of lockdowns per se, but that’s not really discussed in the paper.

3. Dif in Dif Estimation

The real time moment for the Ukraine shock is the same for all countries. But the real time before/after lockdown varies by country by a few weeks. This variance can be important, especially for more granular analysis as in the event study, because the new dif-in-dif literature (which is not referred to in the paper) spends a lot of time discussing how to deal with potential confounding arising from combining different pre- and post-periods in pooled data when the timing of the ‘treatment’ differs across units. This needs thought and probably further empirical investigation.

More broadly there is no discussion of the identification assumptions relating to dif-in-dif and no examination of them, eg. In relation to common trends. There is evidence of some anticipation effects which, again, needs further discussion/examination because that’s problematic for dif-in-dif estimation.

It’s hard to think of 2021 as a ‘counterfactual’ period given the dynamic and on-going effects of COVID. The authors talk of standardising their Tweet sentiment data in a way that they say helps deal with this but I was unable to work out precisely how this standardisation was undertaken and why it would address the issue.

I found some of the analysis difficult to follow. It seems that the authors have high-frequency (daily) data but much of the dif-in-dif is presented as if the unit of analysis is the year. This does not do justice to the fine time-varying data they have and could prove misleading if there is systematic measurement error that this level of aggregation introduces.

As Mark Bryan and Stephen Jenkins pointed out in their seminal paper if one aggregates data to the macro (country) level the analyst has to recognise that the N observations is countries not individuals. This has implications for the adjustment to standard errors. In this case, if I understand the dif-in-dif modelling, the data are micro level with country fixed effects, but of course individuals’ responses are clustered within country. I think the authors need to spend more time explaining how they treat these data for the purposes of standard error adjustments since it may be that the significance of results depends upon this.

Recent work shows just how sensitive some measures of wellbeing are to seasonality. One of these papers is in Plos One (Blanchflower, D. G. and Bryson, A. (2022) ”COVID and Mental Health in America”, PLoS ONE, 17(7): e0269855). It may be worth thinking more about this issue when trying to compare wellbeing across years. It may be that incorporating day/week controls is sufficient but it may be good to double-check.

4. Data

I am not familiar with the GNH metric they construct and, perhaps like many other readers, are disinclined to hunt down other papers to find out. There should be a fulsome data appendix explaining precisely how the data are constructed – including the GNH – which also explains why the metric is appropriate, so that readers can check it out.

The GNH may, in fact, not be the best wellbeing metric since the growing literature on the measurement of wellbeing points to important differences in the determinants of and trends in positive and negative affect. They are asymmetrical in their responses to events such as those in this paper. If one had used a measure of negative affect we might have seen quite a different story. It would be a very nice addition if the authors could add in alternative wellbeing metrics to see whether negative affect responds differently to positive affect.

5. Heterogeneity

The literature on shocks indicates that their impact on wellbeing depletes with distance from the shock in both time and geography. I imagine this is what prompted the North v South investigation in the paper, although I couldn’t see that clearly stated. I was wondering if there was a better way to capture distance from the shock by interacting that distance with the post-period? I didn’t find the North v South analysis particularly illuminating.

6. Magnitude of effects and the policy implications

The authors point to a very short-run dip in wellbeing after lockdown/Ukraine, with something that looks like regression to the mean shortly afterwards. This finding is common in the literature. The authors don’t quantify the effect: just how big was it while it lasted? If it is not that big, and does not last that long, then why should we be concerned about it? This sort of question prompted Angus Deaton in a 2010 Oxford Economics Paper to wonder just how much policy makers needed to pay attention to such effects.

Somewhere one expects discussion of the magnitude of COVID (not lockdown) and Ukraine shocks to wellbeing if this paper is going to make a real contribution.

7. Minor points

- the abstract is overly long and not very informative

- p. 10: I have no idea what NRC and VADER are.

- the figures aren’t easy to read. It would be worth identifying t=0 and before/after periods on the x-axis.

6. PLOS authors have the option to publish the peer review history of their article (what does this mean?). If published, this will include your full peer review and any attached files.

Reviewer #1: No

Reviewer #2: No

---

## [Author Response · Author response to Decision Letter 0]

1 Jun 2023

Please see submitted document 'Response to reviewers'.

---

## [Decision Letter · Decision Letter 1]

26 Jun 2023

PONE-D-22-19460R1Reactions to macro-level shocks and re-examination of adaptation theory using Big DataPLOS ONE

Dear Dr. Rossouw,

Thank you for submitting your manuscript to PLOS ONE. After careful consideration, we feel that it has merit but does not fully meet PLOS ONE’s publication criteria as it currently stands. **While Reviewer 1 recommended accepting the paper for publication, Reviewer 2 had a different opinion. **

**Reviewer 2 acknowledges the authors' efforts in revising the paper in response to initial comments. However, they have expressed concerns regarding the estimation strategy and its implementation, which makes interpreting results difficult. Key issues raised include:**

*
**1. The authors are using 2021 as a counterfactual period for lockdowns occurring in 2020. Reviewer 2 finds it problematic as it's complex to compare treatment in two distinct periods, especially given the ongoing evolution of the COVID-19 pandemic. They question the adequacy of conditioning solely on COVID-19 cases.**
*

*
**2. The reviewer mentions concerns about assumptions of common trends, anticipation effects, and variance in the lockdown treatment that still need to be fully resolved in the current version of the paper.**
*

*
**3. Reviewer 2 also points out that the authors need to sufficiently address the limitations and implications of having a small sample size of only ten countries. **
*

**Overall, the reviewer believes that while some of these issues may be resolvable, significant problems need to be addressed to improve the paper's quality. Therefore, I invite you to take this revise-and-resubmit chance to address Reviewer 2's concerns appended below.**

We look forward to receiving your revised manuscript.

Kind regards,

Ali B. Mahmoud, Ph.D.

Academic Editor

PLOS ONE

Reviewers' comments:

Reviewer's Responses to Questions

**Comments to the Author**

1. If the authors have adequately addressed your comments raised in a previous round of review and you feel that this manuscript is now acceptable for publication, you may indicate that here to bypass the “Comments to the Author” section, enter your conflict of interest statement in the “Confidential to Editor” section, and submit your "Accept" recommendation.

Reviewer #1: All comments have been addressed

Reviewer #2: (No Response)

2. Is the manuscript technically sound, and do the data support the conclusions?

Reviewer #1: Partly

Reviewer #2: Partly

3. Has the statistical analysis been performed appropriately and rigorously? 

Reviewer #1: Yes

Reviewer #2: No

4. Have the authors made all data underlying the findings in their manuscript fully available?

Reviewer #1: No

Reviewer #2: Yes

5. Is the manuscript presented in an intelligible fashion and written in standard English?

Reviewer #1: Yes

Reviewer #2: Yes

6. Review Comments to the Author

Reviewer #1: (No Response)

Reviewer #2: Comments on PONE-D-22-19460r1 “Re-examining adaptation theory using Big Data: Reactions to External Shocks”

The revised paper is clearer than the original in terms of the nature of the shocks it is investigating, the data available and the nature of the analysis. But a number of issues remain which mean the paper is not ready for publication.

1. Macro analysis

Thanks for clarifying the point regarding the macro nature of the data which tracks country-level sentiment and thus provides the basis for cross-country comparison.

The checks on external validity are welcome because I think some will be sceptical about your ability to capture wellbeing for a representative population with twitter sentiment.

2. The shocks

In the revised paper the authors say they are testing responses to two shocks: a war and a ‘government regulation’ shock. The latter isn’t really the shock: it’s a policy response to the COVID shock. This is acknowledged but it’s tricky disentangling the two in that COVID is about the health shock (infection, illness, death) and the government response is lockdown. You control for COVID cases on the right-hand side of the equation. I’m really not sure that’s the right way to go since lockdown is, arguably, jointly determined with COVID cases, notwithstanding some cross-country variance in the policy.

3. Dif in Dif Estimation

I note the authors have consulted the new dif-in-dif literature I referred to on variance in the timing of events for the dif-in-dif across units. This is relevant to the depiction of before and after lockdown since this varies by country by a few weeks. As I said before, this variance can be important, especially for more granular analysis as in the event study, because the new dif-in-dif literature spends a lot of time discussing how to deal with potential confounding arising from combining different pre- and post-periods in pooled data when the timing of the ‘treatment’ differs across units. I can not see how you have accommodated these issues in your analysis and you do not clarify this in your response to reviewers.

I asked for a rendition of whether the dif-in-dif estimates are credible given their identification assumptions, eg. in relation to common trends. However, I didn’t see this addressed in the revision. And the issue of anticipation remains problematic.

4. A counterfactual for lockdown war

Previously I said it was hard to think of 2021 as a ‘counterfactual’ period given the dynamic and on-going effects of COVID. The authors said they had to have a post-COVID period for the counterfactual since COVID created a new ‘normal’. The problem is that it occurs after lockdown and not before, and it was not a stable new normal. So this is potentially really problematic. I can think of no other dif-in-dif estimation that that takes a post-period as the counterfactual.

5. The N observations

The fact that the data are aggregated to country level means we have a country-level analysis. As such the N is low and the authors need to consider the implications of this for their analysis. I suggest the authors go back to the Bryan and Jenkins paper here https://academic.oup.com/esr/article/32/1/3/2404247 to establish whether their results may be unreliable for the reasons discussed.

6. Heterogeneity

I remain unclear as to the value of the North v South analysis. And the paper does not deal with the issue I raised last time, namely the idea that the impact of shocks on wellbeing depletes with distance from the shock in both time and geography. It would be good to see this with respect to Ukraine.

7. Habituation and effect size

The current estimates suggest the size of the wellbeing shock was bigger for lockdown that Ukraine but that the Ukraine effect lasted longer. Why would this be? You also suggest that the Ukraine effect lasted four weeks. This is a long time in the wellbeing literature. I suggest you have regard to the terrorist literature I referred to in my previous comments. It’s worth thinking about these effects relative to what you find.

7. PLOS authors have the option to publish the peer review history of their article (what does this mean?). If published, this will include your full peer review and any attached files.

Reviewer #1: No

Reviewer #2: No

---

## [Author Response · Author response to Decision Letter 1]

8 Aug 2023

Since the response document contains STATA output, it cannot be pasted in the space provided. Please see the attached word document addressing reviewer #2's additional comments.

---

## [Decision Letter · Decision Letter 2]

30 Aug 2023

PONE-D-22-19460R2Reactions to macro-level shocks and re-examination of adaptation theory using Big DataPLOS ONE

Dear Dr. Rossouw,

Thank you for submitting your manuscript to PLOS ONE. After careful consideration, we feel that it has merit but does not fully meet PLOS ONE’s publication criteria as it currently stands. Certainly, **whilst *Reviewer 2* commended your revision efforts, they, however, remained *"concerned about the methodological aspects of the paper, which make interpretation difficult."* **Therefore, we invite you to submit a revised version of the manuscript that addresses the points raised during the review process.

We look forward to receiving your revised manuscript.

Kind regards,

Ali B. Mahmoud, Ph.D.

Academic Editor

PLOS ONE

Reviewers' comments:

Reviewer's Responses to Questions

**Comments to the Author**

1. If the authors have adequately addressed your comments raised in a previous round of review and you feel that this manuscript is now acceptable for publication, you may indicate that here to bypass the “Comments to the Author” section, enter your conflict of interest statement in the “Confidential to Editor” section, and submit your "Accept" recommendation.

Reviewer #2: (No Response)

2. Is the manuscript technically sound, and do the data support the conclusions?

Reviewer #2: Partly

3. Has the statistical analysis been performed appropriately and rigorously? 

Reviewer #2: I Don't Know

4. Have the authors made all data underlying the findings in their manuscript fully available?

Reviewer #2: Yes

5. Is the manuscript presented in an intelligible fashion and written in standard English?

Reviewer #2: Yes

6. Review Comments to the Author

Reviewer #2: The authors have responded in detail to my previous points and have put considerable effort into the paper. I remain concerned about a couple of points where, I suspect, we will have to agree to disagree. It will be for the editor to decide how to handle those differences.

The first is the issue of treating lockdown as a ‘shock’ since it is really COVID that was the shock and the lockdown was a response to that shock. I appreciate that the paper is not the first to do this. Even so, I find it hard to interpret the impact of lockdown with COVID used as independent right-hand side variables.

Second, I think that treating 2021 as a counterfactual period is problematic due to the dynamic and on-going effects of COVID. It’s very tricky interpreting results relative to a counterfactual period which is itself unstable.

Third, I’d like to have seen some engagement with the common trends assumption in the paper but there is none despite the fact that it’s crucial to dif-in-dif estimation. If there is no common trend then one needs to resort to something to tackle that such as synthetic matched dif-in-dif.

Fourth, I still don’t see the value of the North v South analysis. If it’s intended to capture intensity of treatment it’s a pretty crude way of doing so.

Finally I also remain concerned by the N observations (point 5 in my previous comments) notwithstanding the additional test the authors report.

7. PLOS authors have the option to publish the peer review history of their article (what does this mean?). If published, this will include your full peer review and any attached files.

Reviewer #2: No

---

## [Author Response · Author response to Decision Letter 2]

13 Oct 2023

Please see attached Word document.

---

## [Decision Letter · Decision Letter 3]

17 Oct 2023

PONE-D-22-19460R3Reactions to macro-level shocks and re-examination of adaptation theory using Big DataPLOS ONE

Dear Dr. Rossouw,

Thank you for submitting your manuscript to PLOS ONE. After careful consideration, we feel that it has merit but does not fully meet PLOS ONE’s publication criteria as it currently stands. Therefore, we invite you to submit a revised version of the manuscript that addresses the points raised during the review process.

We look forward to receiving your revised manuscript.

Kind regards,

Ali B. Mahmoud, Ph.D.

Academic Editor

PLOS ONE

Journal Requirements:

Reviewers' comments:

Reviewer's Responses to Questions

**Comments to the Author**

1. If the authors have adequately addressed your comments raised in a previous round of review and you feel that this manuscript is now acceptable for publication, you may indicate that here to bypass the “Comments to the Author” section, enter your conflict of interest statement in the “Confidential to Editor” section, and submit your "Accept" recommendation.

Reviewer #2: All comments have been addressed

2. Is the manuscript technically sound, and do the data support the conclusions?

Reviewer #2: Partly

3. Has the statistical analysis been performed appropriately and rigorously? 

Reviewer #2: (No Response)

4. Have the authors made all data underlying the findings in their manuscript fully available?

Reviewer #2: Yes

5. Is the manuscript presented in an intelligible fashion and written in standard English?

Reviewer #2: Yes

6. Review Comments to the Author

Reviewer #2: Please see the attached set of comments contained in my note to you. You need to incorporate some of what was in your response letter within the article itself.

7. PLOS authors have the option to publish the peer review history of their article (what does this mean?). If published, this will include your full peer review and any attached files.

Reviewer #2: No

---

## [Author Response · Author response to Decision Letter 3]

28 Nov 2023

Our response document contains graphs, therefore please refer to the Word document.

---

## [Decision Letter · Decision Letter 4]

4 Dec 2023

Reactions to macro-level shocks and re-examination of adaptation theory using Big Data

PONE-D-22-19460R4

Dear Dr. Rossouw,

We’re pleased to inform you that your manuscript has been judged scientifically suitable for publication and will be formally accepted for publication once it meets all outstanding technical requirements.

Kind regards,

Ali B. Mahmoud, Ph.D.

Academic Editor

PLOS ONE

Additional Editor Comments (optional):

Reviewers' comments:

Reviewer's Responses to Questions

**Comments to the Author**

1. If the authors have adequately addressed your comments raised in a previous round of review and you feel that this manuscript is now acceptable for publication, you may indicate that here to bypass the “Comments to the Author” section, enter your conflict of interest statement in the “Confidential to Editor” section, and submit your "Accept" recommendation.

Reviewer #2: All comments have been addressed

2. Is the manuscript technically sound, and do the data support the conclusions?

Reviewer #2: Yes

3. Has the statistical analysis been performed appropriately and rigorously? 

Reviewer #2: Yes

4. Have the authors made all data underlying the findings in their manuscript fully available?

Reviewer #2: Yes

5. Is the manuscript presented in an intelligible fashion and written in standard English?

Reviewer #2: Yes

6. Review Comments to the Author

Reviewer #2: I commend the authors for their hard work. Although we may not agree on everything the paper is transparent about choices made so the reader can judge the analysis accordingly.

7. PLOS authors have the option to publish the peer review history of their article (what does this mean?). If published, this will include your full peer review and any attached files.

Reviewer #2: **Yes: **Alex Bryson

---

## [Editor Report · Acceptance letter]

4 Jan 2024

PONE-D-22-19460R4 

PLOS ONE

Dear Dr. Rossouw, 

I'm pleased to inform you that your manuscript has been deemed suitable for publication in PLOS ONE. Congratulations! Your manuscript is now being handed over to our production team.

Kind regards, 

on behalf of

Dr. Ali B. Mahmoud 

Academic Editor

PLOS ONE